# Earliest amniote tracks recalibrate the timeline of tetrapod evolution

John A. Long[1], Grzegorz Niedźwiedzki[2,3], Jillian Garvey[4,5], Alice M. Clement[1], Aaron B. Camens[1], Craig A. Eury[6], John Eason[6] & Per E. Ahlberg[2✉]

The known fossil record of crown-group amniotes begins in the late Carboniferous with the sauropsid trackmaker *Notalacerta*[1,2] and the sauropsid body fossil *Hylonomus*[1–4]. The earliest body fossils of crown-group tetrapods are mid-Carboniferous, and the oldest trackways are early Carboniferous[5–7]. This suggests that the tetrapod crown group originated in the earliest Carboniferous (early Tournaisian), with the amniote crown group appearing in the early part of the late Carboniferous. Here we present new trackway data from Australia that challenge this widely accepted timeline. A track-bearing slab from the Snowy Plains Formation of Victoria, Taungurung Country, securely dated to the early Tournaisian[8,9], shows footprints from a crown-group amniote with clawed feet, most probably a primitive sauropsid. This pushes back the likely origin of crown-group amniotes by at least 35–40 million years. We also extend the range of *Notalacerta* into the early Carboniferous. The Australian tracks indicate that the amniote crown-group node cannot be much younger than the Devonian/Carboniferous boundary, and that the tetrapod crown-group node must be located deep within the Devonian; an estimate based on molecular-tree branch lengths suggests an approximate age of early Frasnian for the latter. The implications for the early evolution of tetrapods are profound; all stem-tetrapod and stem-amniote lineages must have originated during the Devonian. It seems that tetrapod evolution proceeded much faster, and the Devonian tetrapod record is much less complete, than has been thought.

The origin of tetrapods, understood as an evolutionary and ecological phenomenon, was not a single event but a process that began with the acquisition of incipient terrestrial locomotory competence in the tetrapod stem group and ended with the emergence of the major crown-group clades, amphibians and amniotes. Of particular importance for the future development of the global ecosystem was the origin of amniotes, the only tetrapod clade to achieve complete reproductive independence of water, and by far the most impactful in terms of both diversity and disparity.

An overall understanding of this phase of vertebrate evolution requires data on phenotypic change, the timing of evolutionary and cladogenetic events, and patterns of diversity, disparity and biogeography. Three principal data sources are available: body fossils, ichnofossils (footprints and other traces) and time-calibrated molecular phylogenetic divergence dates. Body fossils and ichnofossils are typically preserved in different sedimentation regimes, and can thus capture animals with different environmental preferences, but both require taphonomic settings with net sediment deposition rather than erosion, and will thus be biased towards lowland environments, although some upland depositional settings are also known[10]. Molecular divergence dates are unaffected by depositional environments, but are themselves partly dependent on fossil calibration of the phylogeny. Furthermore,

they can date only phylogenetic nodes uniting living lineages, such as the tetrapod crown-group node (uniting the lissamphibian and amniote lineages) and the amniote crown-group node (uniting the mammal and reptile–bird lineages). Fossils, by contrast, can illuminate the details of morphological evolution within stem groups.

Molecular divergence dates for the amniote crown-group node from 30 recent studies (Supplementary Information Part 1), curated at the TimeTree website (https://www.timetree.org), form a tight cluster with a median age of 319 million years, which corresponds to early Bashkirian (mid-Carboniferous); the spread of the cluster is 308.5 to 334.7 million years, thus spanning from Moscovian (late Carboniferous) to Viséan (early Carboniferous). The corresponding date cluster of 32 dates for the tetrapod crown-group node has a much wider spread, ranging from 333.3 to 395.0 million years (that is, from the Viséan to the Eifelian (Middle Devonian)); the median age in this case is 352 million years, or Tournaisian (earliest Carboniferous). The preponderance of molecular evidence thus suggests an origin of the tetrapod crown group during the earliest Carboniferous, with crown amniotes appearing some 30–35 million years later. This places these events in the aftermath of the end-Devonian mass extinction, during and after the 20-million-year interval of poor fossil record known as Romer's gap[11]. The published fossil record is compatible with this time frame, showing the earliest

[1]College of Science and Engineering, Flinders University, Adelaide, South Australia, Australia. [2]Department of Organismal Biology, Uppsala University, Uppsala, Sweden. [3]Polish Geological Institute – National Research Institute, Warsaw, Poland. [4]Department of Archaeology and History, La Trobe University, Bundoora, Victoria, Australia. [5]DJANDAK, Dja Dja Wurrung Enterprises, Bendigo, Victoria, Australia. [6]Independent researcher, Jamieson, Victoria, Australia. ✉e-mail: per.ahlberg@ebc.uu.se

crown-group amniote body fossils and trackways (*Hylonomus* and *Notalacerta*) in the Bashkirian[1–4], the earliest crown-group tetrapod body fossils (for example, *Balanerpeton*) in the late Viséan[5,6], and the earliest crown-group tetrapod trackways (for example, *Batrachichnus* and *Palaeosauropus*) in the mid-Tournaisian[7] (Fig. 1a). However, this compatibility partly reflects the calibration of the molecular trees by known fossils, and is thus not a fully independent verification.

We present here new trackway evidence from Taungurung Country, Victoria, Australia (Figs. 1b,c and 2), indicating that these dates are substantially too late. Crown amniotes were already present in northeast Gondwana by the early Tournaisian. This in turn implies that the crown tetrapod node must lie deep in the Devonian. New trackway data from Silesia in Poland show that the earliest records of crown amniotes in the equatorial regions of Euramerica are also earlier than previously thought, Serpukhovian rather than Bashkirian.

## The Australian tracks

The Australian tracks are preserved on the upper surface of a loose but essentially in situ fine-grained silty sandstone block from the bank of the Broken River at Barjarg, Taungurung Country, Victoria (Museums Victoria specimen NMV P258240). In the Taungurung language, this section of the Broken River was referred to as *Berrepit*, meaning to flee or run away[12]. The block derives from the Home Station Sandstone member of the Snowy Plains Formation in the upper part of the Mansfield Group (Fig. 1b,c). Although aquatic invertebrate and fish trail trace fossils have previously been described from other locations within the Home Station Sandstone[13], this is the first record of terrestrial vertebrate tracks. This new specimen was discovered by the two non-professional members of the author group (C.A.E. and J.E.), who brought it to the attention of the professional palaeontologists, in a demonstration of the value of citizen science. The locality lies within the Lachlan Fold Belt, a tectonically complex region with multiple sedimentary basins and several orogenic episodes during the Palaeozoic (Supplementary Information Part 2). The sedimentary deposits in the region are predominantly Devonian and include several important vertebrate localities[8]. However, the upper Mansfield Group[9] contains a characteristic early Carboniferous vertebrate assemblage without Devonian index taxa such as placoderms, tristichopterids and porolepiforms[14–17]. An associated assemblage of vertebrate microremains has yielded a tooth of the chondrichthyan genus *Ageleodus*, which closely resembles teeth of this genus from the Famennian (latest Devonian) of the Catskill Formation in Pennsylvania, rather than examples from later Carboniferous localities[14]. Deposition in this region was terminated by the Kanimblan orogeny, which began in the latest Devonian and during which active folding seems to have come to an end by the late Tournaisian[18,19] (Supplementary Information Part 2). The Snowy Plains Formation must thus belong to the early part of the Tournaisian; it probably falls within the age span 358.9 to 354 million years old.

The track surface, which is dense, fine-grained and very well preserved, carries three generations of subaerial original surface tracks, preserved in concave epirelief, which all seem to have been made by the same trackmaker taxon (Fig. 2a,b). The oldest is an isolated pes print (Ip). A brief rain shower after this footprint had been made left it, as well as the general surface, pockmarked with raindrop prints. Shortly after the rainfall, while the ground was still moist, a trackway (A) was made by an animal that left well-defined foot impressions. Sometime later, when the ground had begun to dry and harden, another trackway was made (B) that consists largely of well-preserved claw scratches with faint accompanying footprints. Neither trackway is associated with a body or tail drag. The spacing of manus and pes prints in trackway A implies a hip–shoulder distance of approximately 17 cm if the animal was trotting, slightly more if it was performing a sequential walking gait. The total body length is impossible to determine because neck plus head length and tail length are unknown, but applying the proportions of a modern water monitor (*Varanus salvator*), which has a broadly similar foot morphology, gives a suggested length in the region of 80 cm.

The tracks present a consistent foot morphology, well documented by the combined evidence of the different footprints (Supplementary Information Part 3). The manus is smaller than the pes (Fig. 2g). Both are pentadactyl, with five relatively long, slender digits splayed out into a fan shape, although digit V does not always leave a distinct impression. Digits I and V are the shortest, III and IV the longest. The digit impressions bulge distally into slightly swollen tips, but there are no distinct phalangeal pad impressions (Fig. 2e,f). The impression of the skin surface appears smooth without distinct scales, although this may be a preservation effect. Digits I–IV are associated with impressions of fairly large sharp claws, similar in relative size to those of a monitor lizard (*Varanus*), whereas digit V carries a very short claw. The claw prints are sometimes deflected medially at approximately 90° to the long axes of the digits, creating a characteristic inverted J shape (Fig. 2c,d). The claws are discrete structures, clearly distinct from the more proximal parts of the digits, as shown by the co-occurrence of softly rounded digit tips with sharp claw scratches in the same footprint (Fig. 2e,f).

This foot morphology carries a clear phylogenetic signal. Claws are a derived character of crown amniotes and are almost invariably present in this clade. Importantly, they are not present in known stem amniotes; seymouriamorphs, diadectids and limnoscelids all lack claws, as evidenced by their footprints[7,20–25]. As claws must have been present at the amniote crown-group node, it is probable that they originated at the very top of the stem group, but no unambiguous clawed stem amniotes have been discovered. Claws were apparently present in microsaurs[10], which have an uncertain phylogenetic position but may be crown amniotes[6]. Outside the crown amniotes, claws or keratinized toe tips occur sporadically in modern anurans (*Xenopus* and *Hymenochirus*)[26] and salamanders (for example, *Onychodactylus*), and may have been present in some temnospondyls judging by the shape of the terminal phalanges[27]. However, ichnotaxa such as *Batrachichnus* and *Limnopus*, which are attributed to temnospondyls, lack claw impressions[7], and in any case, the presence of five digit-impressions on the manus prints of the Snowy Plains Formation slab rules out a temnospondyl identity for our trackmaker.

Within the amniote crown group, the deepest phylogenetic split is that between Synapsida (stem and crown mammals) and Sauropsida (stem and crown reptiles, including birds). The earliest known synapsid ichnogenus is *Dimetropus*, thought to represent 'pelycosaur'-grade stem mammals, which is first recorded in the Bochum Formation (late Bashkirian) of Germany[24]. The earliest described sauropsid ichnotaxon is *Notalacerta*, which has been described from mid-Bashkirian localities. The broadly similar *Varanopus* and *Dromopus* are slightly younger[2,25]. These tracks have been ascribed to stem reptiles including captorhinids, protorothyrids and basal diapsids[2,7,25],

*Dimetropus* prints are clearly different from those on the Snowy Plains Formation slab, notably in having longer soles with distinct 'heels', less splayed toes and longer, straighter claws (Fig. 3h). By contrast, the Snowy Plains Formation tracks closely resemble *Notalacerta*, *Dromopus* and in particular *Varanopus* (Fig. 3a–c,e–g). All of these ichnotaxa have a short sole, which often leaves no impression, and digit impressions splayed into a fan shape. They are ectaxonic, meaning that the lateral digits are more strongly developed than the medial ones, and they all show claw impressions, which are not just distinctively sharp-pointed but are also commonly deflected medially (towards the body of the animal) from the long axis of the digit print to create J- or L-shaped digit impressions[1,2]. This exact effect is also seen in the Snowy Plains Formation tracks (Fig. 2); we conclude that the toes of all these trackmakers bore similar claws that were clearly offset from the more proximal part of the digit.

Notwithstanding these similarities, the Snowy Plains Formation tracks also bear some resemblance to another ichnotaxon, *Hylopus hardingi*, which is not attributed to crown amniotes. The presumed

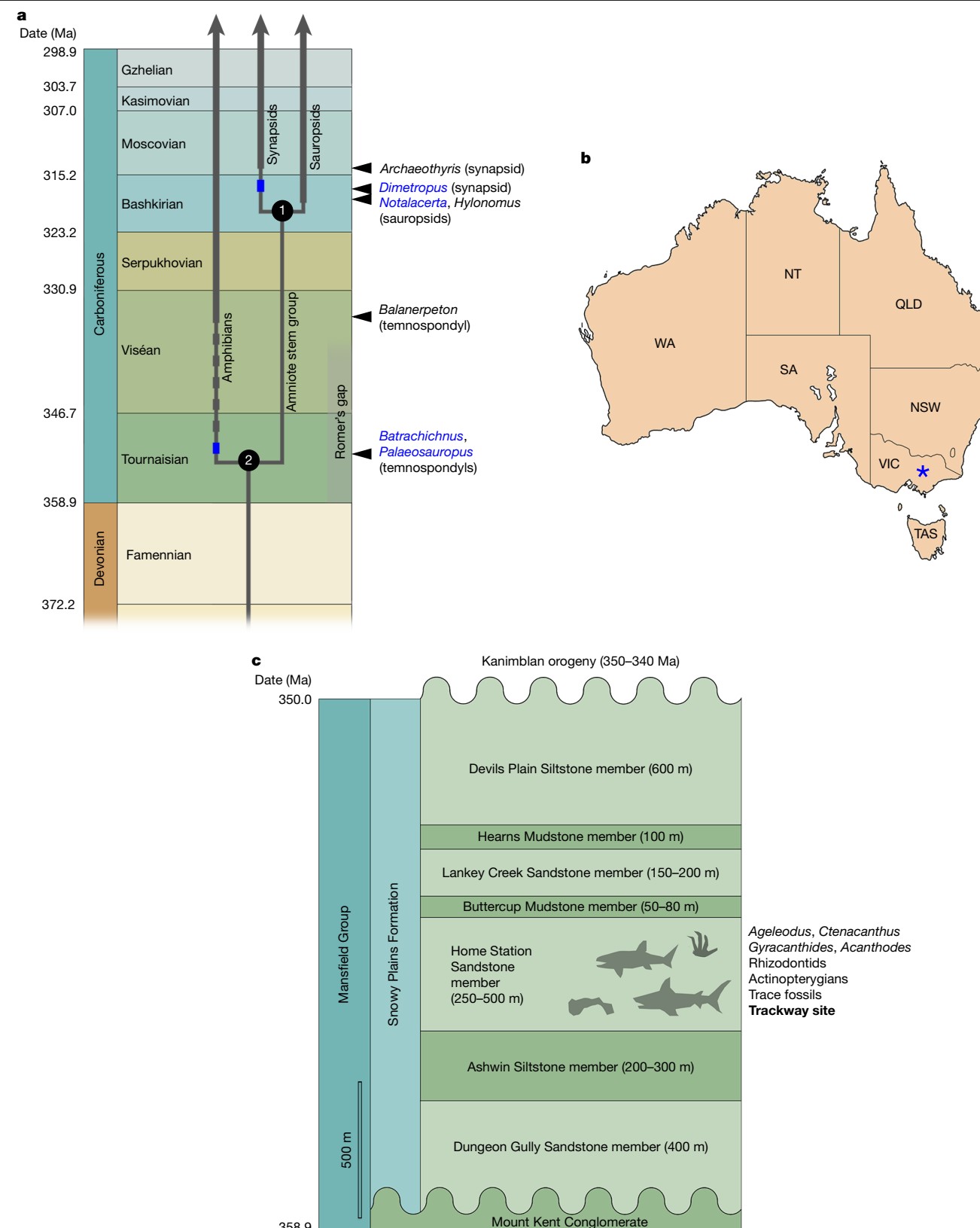

**Fig. 1 | Existing state of knowledge and locality information. a,** Stratigraphic timescale representation of the known early fossil record of crown-group tetrapods. Thin grey lines indicate phylogenetic branches; thick grey lines indicate the body-fossil record from the earliest occurrence; arrowhead and name in black on the right margin indicate the name of the earliest body fossil; blue rectangles indicate the earliest ichnofossil record when this is earlier than the body-fossil record; the dashed line of grey rectangles indicates the range extension between the earliest body fossil and the earliest ichnofossil; name in blue on the right margin indicates the name of the earliest ichnorecord. The amniote crown-group node (1) and tetrapod crown-group node (2) are given minimum ages compatible with the fossil record. All dates are from https://stratigraphy.org/chart. Ma, million years ago. **b,** Map of Australia showing the locality (blue asterisk). NSW, New South Wales; NT, Northern Territory; QLD, Queensland; SA, South Australia; TAS, Tasmania; VIC, Victoria; WA, Western Australia. **c,** Stratigraphy of the Mansfield Group.

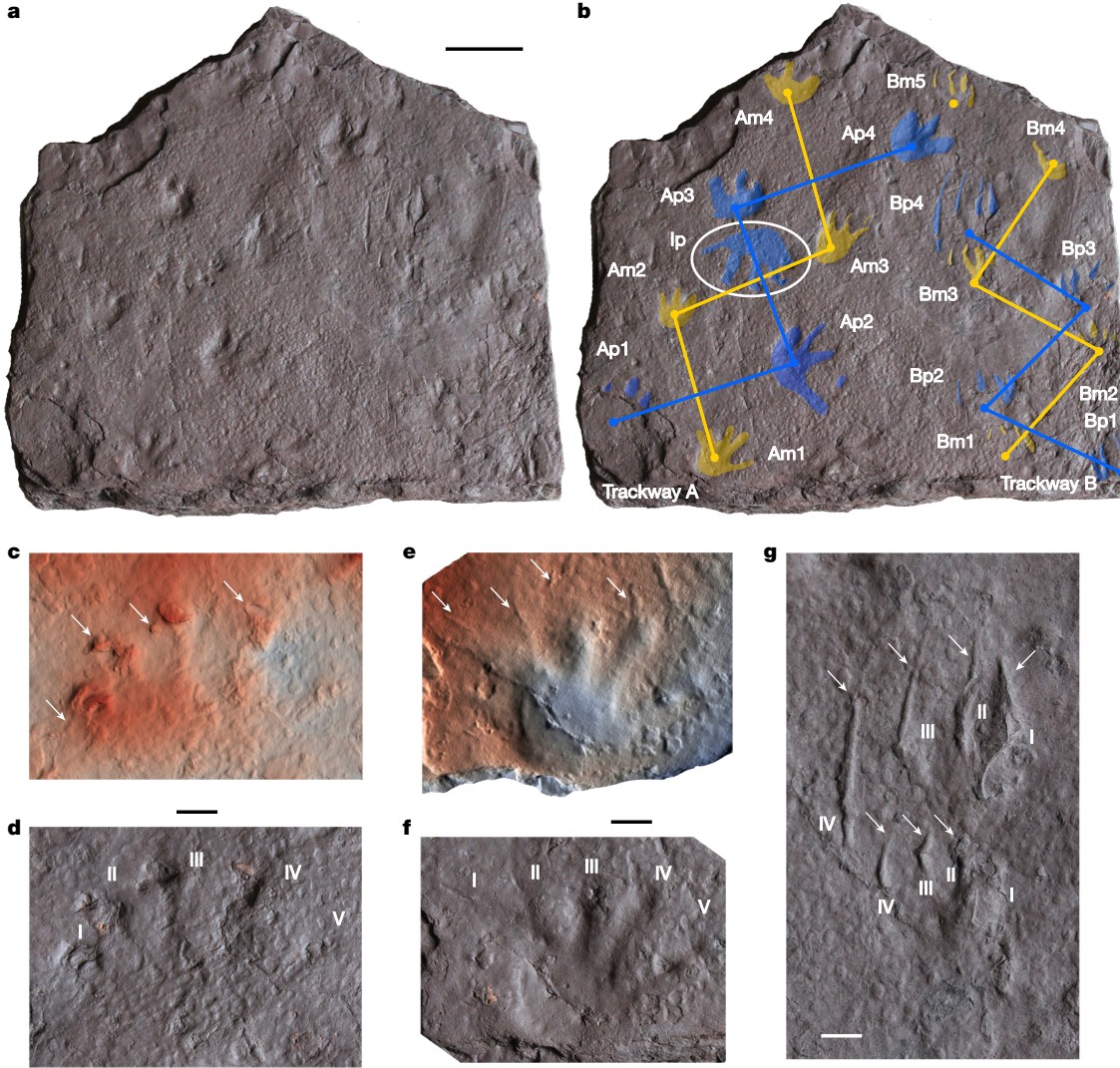

**Fig. 2 | The Snowy Plains Formation trackway slab. a**, Photo of the slab, NMV P258240, as preserved. **b**, Same as in **a**, with footprints and trackways highlighted. Manus (front foot) prints are shown in yellow; pes (hind foot) prints are shown in blue. Am1–4, manus prints from trackway A; Ap1–4, pes prints from trackway A; Bm1–5, manus prints from trackway B; Bp1–4, pes prints from trackway B; Ip, isolated right pes print. **c,d**, Isolated right pes print Ip as a false-colour inverted scan image (**c**) and photo (**d**). **e,f**, Right manus print Am1 as a false-colour scan image (**e**) and photo (**f**). **g**, Photo of pes print Bp4 (above) and manus print Bm3 (below). In **c**–**g**, white arrows denote claw impressions or scratches, Roman numerals denote digit numbers. Scale bars, 50 mm (**a**) and 10 mm (**c**–**g**).

*Hylopus* trackmaker is often referred to as a reptiliomorph[2], an imprecise term that is best interpreted as equivalent to stem amniote, although phylogenetic instability near the tetrapod crown-group node means that several groups of early tetrapods that have been considered as putative *Hylopus* trackmakers are retrieved either as stem amniotes or as stem tetrapods at present depending on the particular analysis[2,6]. Comparison between the Snowy Plains Formation tracks and *Hylopus* is made more complicated by the fact that the tracks attributed to *Hylopus* vary in morphology and include examples that appear to have claw impressions[2], suggesting that some tracks have been misattributed to this ichnotaxon and could themselves represent early unrecognized sauropsids. However, typical and well-preserved *H. hardingi* footprints have distinctive, rounded (almost ball-shaped) toe-tip impressions without any trace of claw marks, and a very short digit V on the manus[2]. The Snowy Plains Formation footprints with their unambiguous discrete claw marks and long digit V on the manus cannot be attributed to *H. hardingi*.

This character distribution allows for two possible phylogenetic interpretations. There is general agreement that *Notalacerta*, *Dromopus*

and *Varanopus* are sauropsid ichnotaxa, but whereas the foot morphology at the amniote crown-group node must have been pentadactyl and claw-bearing, the exact shape of the foot at this node is not known. The overall shape similarity between these ichnotaxa and *H. hardingi* may reflect a foot morphology conserved across the amniote crown-group node. The occurrence of some *Hylopus*-like traits in the Snowy Plains Formation tracks suggests either that the trackmaker was a very primitive sauropsid, phylogenetically basal to the *Varanopus*, *Notalacerta* and *Dromopus* trackmakers, or that it occupied a position close to the amniote crown-group node. In the first case, the crown-group node must predate this trackway slab; in the second, it could be contemporary with it. A substantially younger date for the crown-group node is ruled out, unless the crown amniote characters of the Snowy Plains Formation tracks are dismissed (with no evidential basis) as convergences.

## The Silesian tracks

The earliest crown amniote fossils acknowledged in the current literature are trackways of *Notalacerta* and associated body fossils of

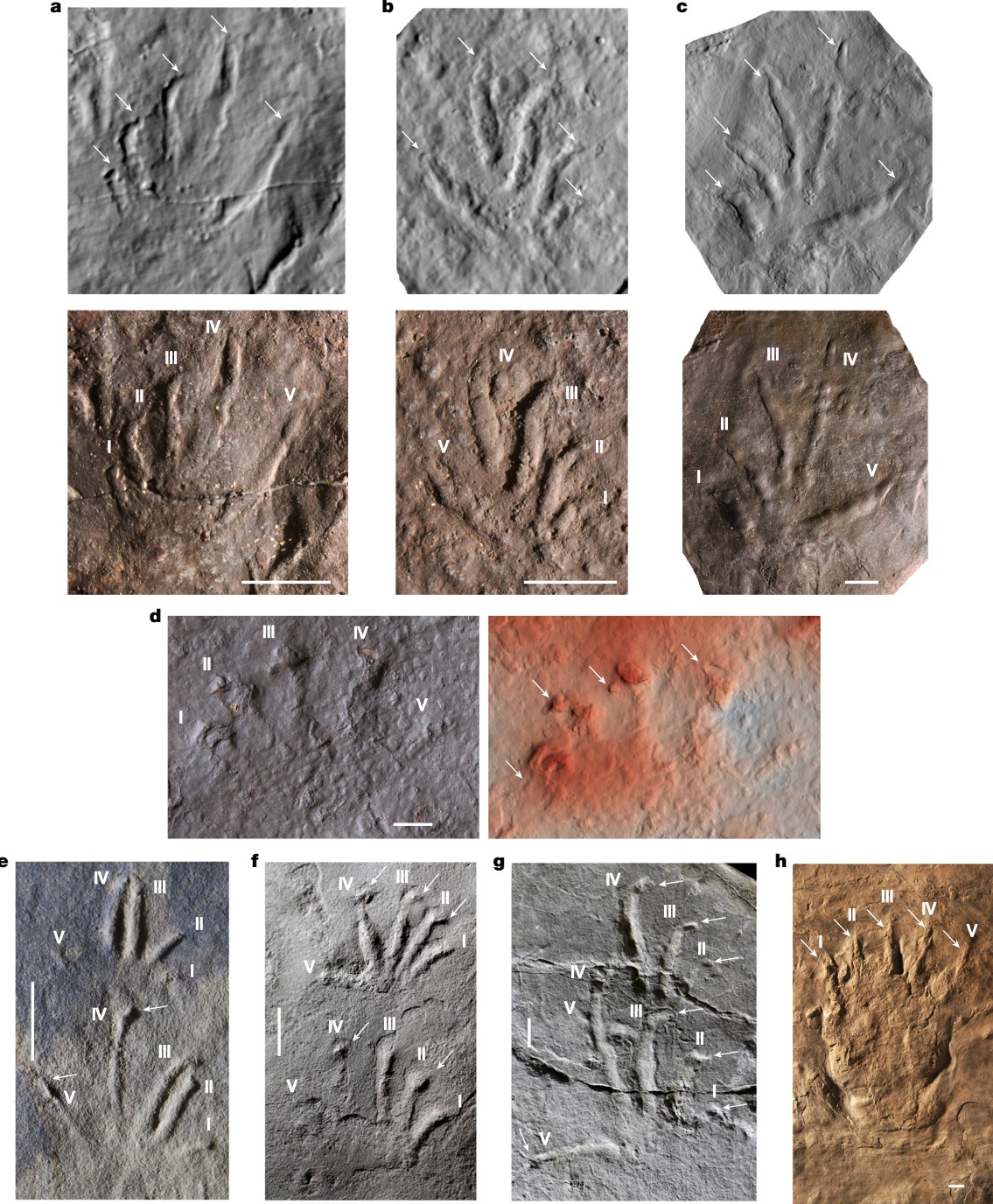

**Fig. 3 | Amniote footprints. a–c**, Three footprints of *Notalacerta* from the middle Serpukhovian to early Bashkirian Wałbrzych Formation of Silesia, Poland; each is shown as an optical scan (top) and photo (bottom). Holy Cross Branch of the Polish Geological Institute – National Research Institute in Kielce, Muz. PGI-OS 220/182 (**a**), 184 (**b**) and 185 (**c**). **d**, Isolated left pes print Ip from the Snowy Plains Formation slab NMV P258240 (Fig. 2c,d), reproduced here to facilitate comparison with other amniote footprints. **e–g**, Presumed sauropsid prints, manus (top) and pes (bottom), of *Notalacerta* (**e**), *Varanopus* (**f**) and *Dromopus* (**g**), all from ref. 2. **h**, *Dimetropus* manus or pes imprint, natural cast, IGWU-1, Geological Museum of the Institute of Geological Sciences, University of Wrocław, Wrocław. Labelling as in Fig. 2. Scale bars, 10 mm. Photos in **e–g** reproduced from ref. 2, Frontiers Media, under a Creative Commons licence CC BY 4.0 (https://creativecommons.org/licenses/by/4.0/).

the stem reptile *Hylonomus* (which may have been the trackmaker) in the middle Bashkirian of Joggins, Nova Scotia, Canada[1,2]. However, during this study, tracks similar to those of *Notalacerta* have also been discovered in the Wałbrzych Formation of the Intra-Sudetic Basin of Silesia in Poland, which has been dated as mid-Serpukhovian to early Bashkirian (Namurian A) on the basis of palynostratigaphy[28] (Fig. 3a–c). This pushes back the amniote record of Euramerica by

approximately eight million years. The Carboniferous–Permian sites of the Intra-Sudetic Basin (Czech Republic and Poland) are historically important for the study of tetrapod tracks and have been studied for more than 150 years[29]. Rich material collected from the Carboniferous part of this succession provides a new guide for resolving the first appearance, diversification and environments of early amniotes in this part of Euramerica (Supplementary Information Part 4).

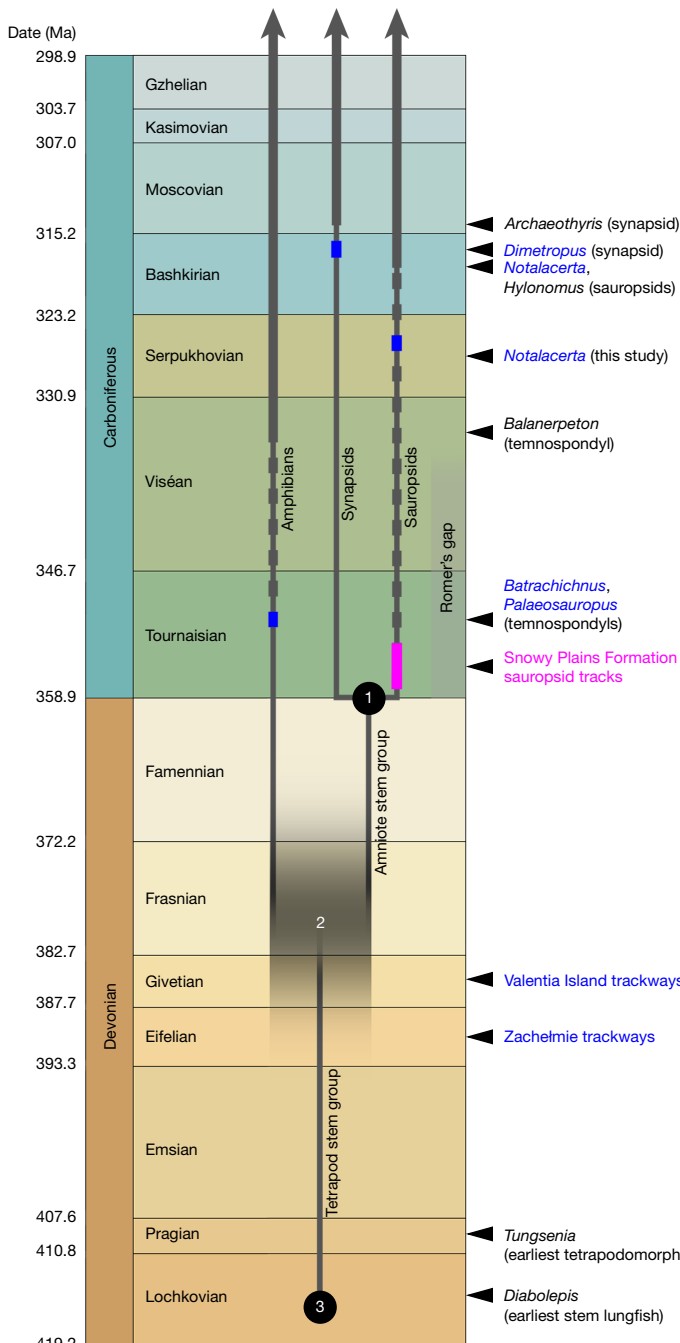

**Fig. 4 | Revised timescale of early tetrapod evolution.** Stratigraphic timescale representation of the Devonian and Carboniferous, showing the impact of the Snowy Plains Formation sauropsid tracks. The track record is shown as a pink rectangle, of double height to indicate possible age range. Other graphic conventions as in Fig. 1. The amniote crown-group node (1) and lungfish–tetrapod node (3) are given minimum ages compatible with the fossil record. The tetrapod crown-group node (2) is positioned in accordance with the branch-length proportions derived from TimeTree (https://www.timetree.org) as explained in the text; vertical blurring of the horizontal branch segment indicates that this date is uncertain and should be considered only as a general indicator, not a precise estimate.

## Discussion

The implications of the Snowy Plains Formation trackways are profound and wide-ranging (Fig. 4). In terms of tetrapod evolution, the Devonian Period has, until now, been seen as the exclusive domain of the stem group. All recent palaeontological analyses place the lungfish–tetrapod node close to the Silurian/Devonian boundary[30–32]. The earliest known fish member of the tetrapod stem group, *Tungsenia*, is Pragian[31]. Limbed stem tetrapods appear in the ichnorecord in the Eifelian, and in the body-fossil record in the Frasnian[33–37]. The known tetrapod body-fossil record of the Frasnian and Famennian consists exclusively of stem-group forms. This fits with the idea of the crown-group radiation as a post-Devonian phenomenon following (and possibly impelled by) the end-Devonian extinction event[10,37]. However, the Snowy Plains Formation trackways challenge this interpretation.

A substantial time interval between the tetrapod and amniote crown-group nodes is a universal feature of recent molecular phylogenies in which both nodes are defined (Supplementary Information Part 1); the median age difference between these nodes in the calibrated phylogenies curated by TimeTree is 33 million years. These age differences are inferred from branch lengths that in turn reflect base substitutions recorded in the genomes of extant animals, and are thus not susceptible to the distorting effects of gaps in the fossil record. Even though the exact ages vary between phylogenies, depending both on the fossil calibrations and phylogenetic algorithms used, the substantial age separation between the amniote and tetrapod crown-group nodes is consistent and must be real; these cladogenetic events were separated by tens of millions of years.

As the Snowy Plains Formation is early Tournaisian in age, a sauropsid identification of the tracks implies that the amniote crown-group node can, at a minimum, be only marginally younger than the Devonian/Carboniferous boundary. This in turn means that the tetrapod crown-group node must lie much further back in the Devonian. To arrive at a rough age estimate for the tetrapod crown-group node, one avenue is to look at the lungfish–tetrapod node, which marks the bottom end of the tetrapod stem group. The median TimeTree age for this node is 408 million years. The corresponding ages of the amniote and tetrapod crown-group nodes are 319 and 352 million years; if the time distance from the amniote crown-group node to the lungfish–tetrapod node is given the unit value 1, the corresponding distance from the amniote crown-group node to the tetrapod crown-group node is 0.371, and that from the tetrapod crown-group node to the lungfish–tetrapod node is 0.629.

In fact, the earliest unambiguous stem lungfish, *Diabolepis*, is approximately 415 million years old (Xitun Formation, Lochkovian, China)[38], so the inferred date for the lungfish–tetrapod node is slightly too young. If, as a thought experiment, the amniote crown-group node and the lungfish–tetrapod node are fixed, respectively, to the Devonian/Carboniferous boundary (358.9 million years) and the mid-Lochkovian (415 million years), and the aforementioned relative branch lengths are applied, they place the tetrapod crown-group node at a median age of 379.7 million years (early Frasnian). This should be understood only as the approximate mid-point of a wide zone of possibility (Fig. 4). However, a much younger age, at or close to the Devonian/Carboniferous boundary, can be rejected because the internode to the amniote crown-group node becomes implausibly short and incompatible with the substantial branch lengths consistently recovered by molecular phylogenies (Supplementary Information Part 1). Conversely, as neither the amniote crown-group node nor the lungfish–tetrapod node has a constrained maximum age, all three nodes could in fact be considerably older than indicated. The rapidly expanding number of sequenced vertebrate genomes creates potential for more robust future phylogenetic analyses that, with the inclusion of the Snowy Plains Formation tracks as a calibration point, can provide a more precise estimate of the tetrapod crown-group node date.

The earliest fossils of limbed tetrapods, the trackways from Zachełmie in Poland (Eifelian)[29] and Valentia Island in Ireland (Givetian)[30], are, respectively, about 390 and 385 million years old, and thus compatible with this new inferred timeline (Fig. 4). However, the widely accepted picture of Devonian tetrapods as a low-diversity array of primitive fish-like forms[10,37] must be false. The cladogenetic

event that gave rise to the tetrapod crown group was preceded by a series of others that gave off the various clades of limbed stem tetrapods, such as baphetids, colosteids and ichthyostegids, and before that the elpistostegalians and various tetrapodomorph fishes[6,37]. All of these cladogenetic events must now be fitted into, approximately, the first two-thirds of the Devonian period. The origins of stem amniote lineages such as seymouriamorphs and diadectomorphs must lie in the Late Devonian. Remarkably, the inferred age of the tetrapod crown-group node presented here is approximately contemporary with the elpistostegalians *Elpistostege* and *Tiktaalik*, often perceived as antecedents and potential ancestors of tetrapods[39–42]. This result strongly supports the much earlier origin of limbed tetrapods indicated by the Middle Devonian trackway record, and implies that tetrapods underwent a far faster process of cladogenesis and morphological evolution during the Devonian than has hitherto been recognized.

A series of body-fossil discoveries over the past four decades lend indirect support to this contention, by providing evidence for previously unsuspected diversity and morphological disparity among Devonian tetrapods[43–47]. Particularly noteworthy is the fact that each new tetrapod locality has yielded one or more new tetrapods, a marked contrast with the wide distribution of associated fishes such as *Holoptychius* and *Bothriolepis*, and a sign that our discoveries are sampling a high-diversity global tetrapod fauna with small geographic ranges for individual genera. Nevertheless, the complete absence so far in the Devonian body-fossil record of any crown-group tetrapods, or crownward stem-group clades such as colosteids and baphetids, indicates that this record markedly under-samples the living diversity.

The trackway record casts some additional light on this phenomenon. At present, the oldest records of amphibians, synapsids, sauropsids and limbed stem tetrapods are all ichnorecords[7,24,33,34] (Fig. 4). It is well known from later parts of the vertebrate fossil record that trackway assemblages often capture taxa that are not seen in associated body-fossil assemblages[48], and this also applies to the Devonian and Carboniferous record. The earliest known high-diversity tetrapod trackway assemblage, from the mid-Tournaisian of Blue Beach, Canada, contains taxa that are not represented among the associated body fossils (for example, temnospondyls)[7]. The Mansfield Group contains no known tetrapod body fossils[14–17]. This is also the case for the two Middle Devonian formations that contain published tetrapod tracks, the Givetian Valentia Slate Formation of Valentia Island, Ireland, and the Eifelian Wojciechowice Formation of Zachełmie, Poland; the former yields only fish[49,50], the latter no body fossils at all. The trackway record thus provides direct evidence of the incompleteness of the body-fossil record, and in turn has a key part to play in fleshing out the picture of early tetrapod diversity, even though it is also quite meagre.

The poor fossil record hampers the search for temporal and spatial patterns of distribution. With the discovery of the Snowy Plains Formation tracks, the crown-group amniote record of northeastern Gondwana now predates that of Euramerica by about 30 million years, but we cannot rule out that earlier representatives may eventually be found in Euramerica as well. The mid- to late-Serpukhovian amniote tracks from Silesia are morphologically advanced, with pronounced claws and narrow, elongated digits, and some are quite large (Fig. 3c); this strongly suggests that the evolutionary history of this group is nested deeper in time but not yet recognized in this region (Supplementary Information Part 4). With regard to biogeography and living environments, recent palaeomagnetic reanalysis of the Devonian-Carboniferous pole path of Australia[51,52] has revealed that the continent was located much further north during the Famennian to Viséan than had previously been thought. At the time of deposition of the Mansfield Group, the trackway locality lay at a latitude of approximately 17° south, at the southern edge of the tropics. This is quite similar to the equatorial latitude of the Euramerican *Notalacerta* localities, and does not present a strong case for a distinction between temperate and tropical faunas being a factor in early amniote distribution.

By contrast, the Snowy Plains Formation trackways do cast substantial new light on the effect of the end-Devonian mass extinction event on tetrapod evolution. Until recently[53–55], the tetrapod fossil record showed a hiatus of approximately 20 million years, known as Romer's gap, between the end-Famennian and the late Viséan. The pre-gap and post-gap tetrapods appeared substantially different in character, with the post-gap forms showing much higher diversity and disparity, as well as being more advanced and including crown-group tetrapods in their ranks[10,37]. This gave rise to the idea that the extinction event had served as a reset for tetrapod evolution, allowing the emergence of more modern groups. It has also been linked to terrestrialization after a supposedly aquatic phase of tetrapod evolution in the Devonian[10,37]. This somewhat simplistic conception of Romer's gap and its relationship to tetrapod evolutionary history can now be replaced by a more nuanced interpretation. The presence of sauropsid tracks in the early Tournaisian implies that the tetrapod crown-group radiation was well under way in the Late Devonian, and that not only lineages such as temnospondyls, seymouriamorphs and diadectomorphs but also crownward stem tetrapods such as baphetids and colosteids crossed the Devonian/Carboniferous boundary. If this is correct, the mass extinction did not have a role in the emergence of these derived lineages, although it is still possible that the amniote crown group arose in its immediate aftermath. The impact of the extinction on diversity, and especially on the selective removal of archaic tetrapod lineages, is harder to assess but may have been substantial. With the exception of *Tulerpeton*[56] and *Brittagnathus*[47] in the Devonian, and some possible Devonian-grade tetrapods in the Tournaisian[53–55], all known Devonian tetrapods seem to represent a less crownward segment of the tetrapod stem than any post-Devonian forms[6]. This suggests a selective extinction with appreciable effects on ecosystem structure.

The Snowy Plains Formation trackways have a disproportionate impact on our understanding of early tetrapod evolution because of their combination of diagnostic amniote characteristics and early, securely constrained date. They demonstrate, once more, the extraordinary importance of happenstance and serendipity in the study of severely under-sampled parts of the fossil record. Against this background, two things stand out: first, that the interpretation of such a fossil record is critically dependent on phylogenetic inferences and cannot be 'read' as a literal account of the history of a group; and second, the fundamental, continuing importance of palaeontological fieldwork as a source of new knowledge.

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

## Methods

Specimens were photographed under oblique lighting to emphasize the footprints. Optical scans were undertaken with a RangeVision Spectrum and the resulting STL files were rendered in RangeVision 3D studio 2022.1 for greyscale images or ParaView 5.10.1 for false-colour height maps.

### Inclusion and ethics

This paper describes Australian and Polish fossil material, deposited with public museums in those countries (Museums Victoria; Holy Cross Branch of the Polish Geological Institute – National Research Institute in Kielce; Geological Museum of the Institute of Geological Sciences, University of Wrocław). The authors include Australian (J.A.L., J.G., A.M.C. and A.B.C.) and Polish (G.N.) researchers, as well as the two discoverers of the Australian trackway slab (C.A.E. and J.E.). As the Australian specimen comes from Taungurung Country, we have consulted with Taungurung Elder and language specialist Aunty L. Padgham about the project; she gave us permission to use the Taungurung name for the section of the river where this fossil was located (see also Acknowledgements).

### Reporting summary

Further information on research design is available in the Nature Portfolio Reporting Summary linked to this article.

## Data availability

Optical surface scans of the footprints shown in Figs. 2c,e and 3a–d can be downloaded via figshare at https://doi.org/10.6084/m9.figshare. 25869367 (ref. 57).

57. Long, A. J. et al. Reptile tracks from the earliest Carboniferous of Australia recalibrate the timeline of tetrapod evolution. *figshare* https://doi.org/10.6084/m9.figshare.25869367 (2025).

**Acknowledgements** We acknowledge that NMV P258240 comes from Taungurung Country, and pay our respects to Taungurung Elders past and present, and all of the Taungurung community. We thank Aunty L. Padgham, Taungurung Elder and language specialist, for providing the Taungurung name for the section of the river where this fossil was located; and A. Gołasa, A. Miziołek and P. Menducki for full access to material collected by them from the Sudetic area in Poland, and also for permission to document of the most important specimens from these collections. P.E.A. acknowledges the support of ERC Advanced Grant ERC-2020-ADG 10101963 "Tetrapod Origin". J.A.L. and A.M.C. receive funding from the Australian Research Council, grants DP 220100825 and DP 200103398.

**Author contributions** C.A.E. and J.E. discovered specimen NMV P258240. J.A.L. provided overall project leadership and wrote most of the first draft manuscript. A.M.C. and A.B.C. made early interpretations of the specimen and contributed to the first draft manuscript. G.N. made the final interpretation of NMV P258240, identified the *Notalacerta* footprints from the Wałbrzych Formation, and made all photographs and scan images. J.G. contributed expertise on the Mansfield Group ichnofossils and liaised with the Taungurung community. A.B.C. made footprint and track measurements on NMV P258240 for Supplementary Information Part 3, which he wrote together with J.A.L. and A.M.C. P.E.A. composed all figures and wrote the final draft manuscript excluding Supplementary Information Part 3. All authors commented on the final draft and made modifications to it.

**Funding** Open access funding provided by Uppsala University.

**Competing interests** The authors declare no competing interests.

**Additional information**
**Correspondence and requests for materials** should be addressed to Per E. Ahlberg.

# Reporting Summary

## Statistics

For all statistical analyses, confirm that the following items are present in the figure legend, table legend, main text, or Methods section.

| n/a | Confirmed | |
|---|---|---|
| ☒ | ☐ | The exact sample size (*n*) for each experimental group/condition, given as a discrete number and unit of measurement |
| ☒ | ☐ | A statement on whether measurements were taken from distinct samples or whether the same sample was measured repeatedly |
| ☒ | ☐ | The statistical test(s) used AND whether they are one- or two-sided<br>*Only common tests should be described solely by name; describe more complex techniques in the Methods section.* |
| ☒ | ☐ | A description of all covariates tested |
| ☒ | ☐ | A description of any assumptions or corrections, such as tests of normality and adjustment for multiple comparisons |
| ☒ | ☐ | A full description of the statistical parameters including central tendency (e.g. means) or other basic estimates (e.g. regression coefficient) AND variation (e.g. standard deviation) or associated estimates of uncertainty (e.g. confidence intervals) |
| ☒ | ☐ | For null hypothesis testing, the test statistic (e.g. *F*, *t*, *r*) with confidence intervals, effect sizes, degrees of freedom and *P* value noted<br>*Give P values as exact values whenever suitable.* |
| ☒ | ☐ | For Bayesian analysis, information on the choice of priors and Markov chain Monte Carlo settings |
| ☒ | ☐ | For hierarchical and complex designs, identification of the appropriate level for tests and full reporting of outcomes |
| ☒ | ☐ | Estimates of effect sizes (e.g. Cohen's *d*, Pearson's *r*), indicating how they were calculated |

*Our web collection on statistics for biologists contains articles on many of the points above.*

## Software and code

Policy information about availability of computer code

| Data collection | None |
|---|---|
| Data analysis | RangeVision 3D studio 2022.1, ParaView 5.10.1 |

For manuscripts utilizing custom algorithms or software that are central to the research but not yet described in published literature, software must be made available to editors and reviewers. We strongly encourage code deposition in a community repository (e.g. GitHub). See the Nature Portfolio guidelines for submitting code & software for further information.

## Data

Policy information about availability of data

All manuscripts must include a data availability statement. This statement should provide the following information, where applicable:
- Accession codes, unique identifiers, or web links for publicly available datasets
- A description of any restrictions on data availability
- For clinical datasets or third party data, please ensure that the statement adheres to our policy

All specimens are housed in accredited institutions and have unique identifying collection numbers (shown in the figure legends of Figs. 2 and 3), allowing them to be retrieved by other workers. All stls from optical scans of footprints shown in the manuscript figures have been uploaded to Figshare. The link to the folder containing all the files is https://doi.org/10.6084/m9.figshare.25869367. The Timetree database can be found at https://timetree.org

## Research involving human participants, their data, or biological material

Policy information about studies with human participants or human data. See also policy information about sex, gender (identity/presentation), and sexual orientation and race, ethnicity and racism.

| | |
|---|---|
| Reporting on sex and gender | n/a |
| Reporting on race, ethnicity, or other socially relevant groupings | n/a |
| Population characteristics | n/a |
| Recruitment | n/a |
| Ethics oversight | n/a |

Note that full information on the approval of the study protocol must also be provided in the manuscript.

## Field-specific reporting

Please select the one below that is the best fit for your research. If you are not sure, read the appropriate sections before making your selection.

☐ Life sciences          ☐ Behavioural & social sciences          ☒ Ecological, evolutionary & environmental sciences

For a reference copy of the document with all sections, see nature.com/documents/nr-reporting-summary-flat.pdf

## Ecological, evolutionary & environmental sciences study design

All studies must disclose on these points even when the disclosure is negative.

| | |
|---|---|
| Study description | Stone slabs carrying fossil footprints of early reptiles from the Carboniferous period were examined, the track-makers were identified as far as possible through comparison with previously described and named fossil footprints, and conclusions were drawn from these comparisons. |
| Research sample | One slab from Australia carrying two trackways and a single isolated footprint. This is the only known specimen from its locality and age. Additional material was described from a Polish museum collection. |
| Sampling strategy | On the Australian slab we sampled every footprint. In the Polish material, we picked out enough specimens (three were figured) to be able to make a secure taxonomic determination. |
| Data collection | Data were collected by means of photography and optical surface scanning. Both were performed by Grzegorz Niedzwiedzki. |
| Timing and spatial scale | n/a. The objects being studied (rocks with fossil footprints) are completely static and the timing of data collection has no effect on the study. |
| Data exclusions | No data exclusions |
| Reproducibility | The fossil specimens are all housed in accredited institutions and have unique accession numbers which are given in the paper. They can thus be retrieved and restudied by other researchers without difficulty. |
| Randomization | This is not relevant to the study as no statistical analyses requiring randomization were performed. |
| Blinding | This is not relevant to the study as no analyses requiring blinding were performed. |

Did the study involve field work?          ☒ Yes          ☐ No

### Field work, collection and transport

| | |
|---|---|
| Field conditions | The fieldwork consisted of field-walking the banks of the Broken River near Barjarg, Victoria, Australia, looking for exposed bedrock blocks (sandstones of the Snowy Plains Formation) containing fossils. |
| Location | Banks of the Broken River near Barjarg, Victoria, Australia. |
| Access & import/export | Access was by permission of the landowner; Craig Eury and James Eason (the collectors) remain in touch with him and keep him |

| Access & import/export | informed about the fate of the slab. The slab has neither been imported nor exported, but remains in Victoria. |
| Disturbance | None. The slab was loose and exposed on the surface, it could simply be picked up. |

# Reporting for specific materials, systems and methods

We require information from authors about some types of materials, experimental systems and methods used in many studies. Here, indicate whether each material, system or method listed is relevant to your study. If you are not sure if a list item applies to your research, read the appropriate section before selecting a response.

## Materials & experimental systems

| n/a | Involved in the study |
|-----|-----------------------|
| ☒ ☐ | Antibodies |
| ☒ ☐ | Eukaryotic cell lines |
| ☐ ☒ | Palaeontology and archaeology |
| ☒ ☐ | Animals and other organisms |
| ☒ ☐ | Clinical data |
| ☒ ☐ | Dual use research of concern |
| ☒ ☐ | Plants |

## Methods

| n/a | Involved in the study |
|-----|-----------------------|
| ☒ ☐ | ChIP-seq |
| ☒ ☐ | Flow cytometry |
| ☒ ☐ | MRI-based neuroimaging |

## Palaeontology and Archaeology

| Specimen provenance | Permission was given verbally by the landowner. The specimen has not been exported or sold. |
| Specimen deposition | The specimen has been deposited with Museums Victoria in Melbourne, Victoria, Australia. |
| Dating methods | New dates were not obtained. The Snowy Plains Formation (part of the Mansfield Group) is dated by a combination of index fossils and regional tectonic history. |

☐ Tick this box to confirm that the raw and calibrated dates are available in the paper or in Supplementary Information.

| Ethics oversight | No ethical approval or guidance was sought or required, essentially because this was a chance discovery within a 'citizen science' context. However, as detailed in our separate ethics statement, close contact was kept with the Taungurung aboriginal community, the recognised Traditional Owners of the area where the slab was found, throughout the project. We ensured that use of Taungurung place names and crediting of the community in the paper complied with their wishes. |

Note that full information on the approval of the study protocol must also be provided in the manuscript.

## Plants

| Seed stocks | *Report on the source of all seed stocks or other plant material used. If applicable, state the seed stock centre and catalogue number. If plant specimens were collected from the field, describe the collection location, date and sampling procedures.* |
| Novel plant genotypes | *Describe the methods by which all novel plant genotypes were produced. This includes those generated by transgenic approaches, gene editing, chemical/radiation-based mutagenesis and hybridization. For transgenic lines, describe the transformation method, the number of independent lines analyzed and the generation upon which experiments were performed. For gene-edited lines, describe the editor used, the endogenous sequence targeted for editing, the targeting guide RNA sequence (if applicable) and how the editor was applied.* |
| Authentication | *Describe any authentication procedures for each seed stock used or novel genotype generated. Describe any experiments used to assess the effect of a mutation and, where applicable, how potential secondary effects (e.g. second site T-DNA insertions, mosiacism, off-target gene editing) were examined.* |

