## [Peer Review File · Nature]

Earliest amniote tracks recalibrate the timeline of tetrapod evolution

Corresponding Author: Professor Per Ahlberg

Version 0:

Reviewer comments:

Referee #1

(Remarks to the Author)

This is an interesting MS on a set of trackways discovered in Australia, and the implications of those trackways for our understanding of tetrapod evolution if correctly identified. It is original in its presented data, and extremely important if correct. Data and methodology are correctly presented. Statistics were not applied in the paper.

It seems to me the critical point upon which everything hangs is the date of the trackway. Its somewhat concerning the blocks upon which the tracks were found was loose. The authors pass this off as "essentially in situ" which makes me scratch my head a little. What is the degree of offset of this block from the main intact bed? Millimeters? Centimeters? Meters? Does this block fit into place on the main bedrock?

That the area is structurally complex makes dating this slab fraught with difficulty. There are no radiometric dates. Argumentation, elaborated upon to a degree in Supplemental Information 2, is relative based on two factors: presence or absence of key taxa, and an assumption of the source of sedimentation. If correct that the source of sedimentation is the Kanimblan Orogeny, if it was of short duration, as argued by others, and if dating in the basin in which the trackways were found is the same as in the basin with similar lithologies that could be dated, then the date is constrained to be between the Devonian-Carboniferous Boundary and the end of the Tournaisian. How the authors settled on a date of the earliest Tournaisian (which would further make this locality unique) I have no strong idea. But that is a long list of nested assumptions and it is upon this that everything else hangs--which is a range extension of amniote trackways back in time by 20 odd million years. I would prefer the authors be a bit more circumspect with respect to the real uncertainty that is being passed over, because doing so doesn't fundamentally change the nature of their analysis for the impact on tetrapod evolution. On the other hand, should this slab be from a younger depositional event, then the impact of the paper decreases proportionately to the shortening of the possible range extension.

Relatedly, I pose a question to the authors: can a similar analysis be performed that rules out a date of late Visean to early Serpukhovian can be ruled out if the identification of the sediment source (uplift due to the Kanimblan Orogeny) doesn't hold? Are there deposits of those ages within this "structurally complex region" present? I think their argument would be bolstered if no deposits of this age or especially younger (say, Visean) were anywhere close to the discovery locality of the specimen. A lot is described about older sediments, in other words, but not much about younger. Please fill out the stratigraphic profile to help the reader agree with your estimates.

One other point that can affect date constraints for the discussion of tetrapod evolution is over the validity of the Zachełmie trackways, which is disputed by some authors (Lucas 2015). This is a more minor point because that critic accepts the Valentia trackways, so its a slight younger adjustment of the origination minimum age estimate. I accept that some of the authors are deeply involved in this debate but this point passes without even a mention; I think it should warrant at least a sentence or two of how that would shift estimates of ancient tetrapod origins.

On the overall conclusion of the paper: I think the evidence has been building for some time that the tetrapod fossil record is missing considerable representation, with recent first occurrences for colosteids and crassigyrinds being dragged down in the record some 20-30 million years into the earliest Carboniferous and possibly into the Devonian for the former. Are we

sampling shifting facies not overall tetrapod evolution? Is this why these amniote prints are appearing in Gondwana earlier than seen in Laurasia as argued by the authors? All good points to get into the literature and the field as a whole discussing more actively.

To improve the presentation of the paper: I think my biggest recommendation to the authors is to be very very careful with their language around "early" and "late", because it can be confusing to a general audience when what is meant shifts depending on perspective. One thing about time from the recent back, but about rocks from the bottom up, and what is meant by the use of modifiers can suggest to the reader the opposite of what is meant by the authors. For example, on line 173, after having discussed amniotes originating in both trackways and body fossils in the Serpukhovian, the authors say their new trackways show that "these dates are substantially too late". It caused me, a fellow paleontologist, some pause to figure out what was meant was that in the rock record those old ideas about amniote origins postdate the implied dates by the new fossils (i.e. thinking about data provided by ROCKS), which are from closer to the Devonian. When I first read it though, I was thinking about TIME since dates were being discussed, and a late date would be OLDER (i.e., further back in time). I suggest in that instance to consider rephrasing to "[those dates are] substantially too recent" or "substantial underestimates". Especially for the generalist audience of Nature you should triple check all of these instances to ensure what is clear in your mind is being communicated to folks who don't live and breathe both time and rocks like we paleontologists do (for good or ill).

Normally at this point I would pass the rest of the minor needed edits to a marked up MS copy (which I still attach) but Nature wants for everything to be written out here so, for your edification, the fine points:

- Line 176: strike the comma after Poland
- Line 176: data show (data is plural)
- Line 177: earlier vs older. Do they occur closer to recent or further back in time?
- Line 222-223: a pedantic point--you aren't looking at skin, but skin impressions into substrate. Substrate can lack resolution to preserve finer details of soft tissues.
- Line 230-231: *Xenopus* has claws. These trackways could of course be made by a yet unknown group that acquired amniote-like claws in parallel to amniotes. I understand the authors are implicitly making an argument by parsimony here but since one of the points of the paper is just how poorly taxa are represented by body fossils at this time in the rock record missing out on a group is certainly conceivable.

No further comments. The paper is very well written and thought provoking; well done!

Referee #2

(Remarks to the Author)

This work reports, describes and interprets the first discovery of tetrapod footprints from the Snowy Plains Formation of Australia. It is unquestionably a new fossil record of tetrapod footprints. The material is well-illustrated, although the footprint outline is not well-traced and some key features such as a false-color scale bar for the 3D models are not provided. In the description, some important features such as the preservation as convex hyporelief or concave epirelief are not specified. Also, the specimen is described as sandstone although the surface is clearly finer. Most important, these footprints are compared to the earliest ichnotaxa attributed to amniotes such as *Notalacerta*, *Dromopus*, *Varanopus* and *Dimetropus* and a similarity with *Varanopus* is noticed. However, this work surprisingly does not discuss possible differences and similarities of the material with *Hylopus*, the most common anamniote tracks of the Mississippian. Material currently assigned to this ichnogenus in some work was mistaken for a reptile track (e.g., Falcon-Lang et al., 2007, 2010) and subsequently consistently and thoroughly attributed to anamniotes (e.g., Keighley et al., 2008; Fillmore et al., 2012; Lucas, 2019; Marchetti et al., 2019, 2021). The described material can be confidently assigned to *Hylopus hardingi*, based on a well-preserved pes (Fig. 2b-c) and manus (Fig. 2e-f). Diagnostic features include, among others, a well-impressed basal pad of digit I in the pes, a clearly more ectaxonic manus imprint compared to the pes, a relatively long digit V in the pes, the non-overlapping digit bases. Trackway B can only tentatively assigned to *Hylopus* because of its poor preservation due to incompleteness and digit tip dragging. The purported claw traces are mostly digit tip drag traces, which occur often in the Palaeozoic anamniote track record and do not imply the presence of claws (e.g., Lucas et al., 2022). Also, the potential presence of pointy digit tips/claws in early anamniotes cannot be excluded, also considering the occurrence of claws in some modern anamniotes (e.g., Maddin et al. 2007). Moreover, the comparison with *Varanopus* is made based on outdated literature (e.g., Haubold and Lucas, 2003, Voigt, 2005). *Varanopus* is clearly different from the described material because of the clearly overlapping digit bases, the more ectaxonic pes imprint, the concave proximal sole/palm margin and the possible occurrence of claw traces. Also, the isolated footprints from Poland, of which the locality is not provided, are assignable to *Hylopus hardingi* manual imprints because of their marked ectaxony and overall digit proportions and morphology. Finally, the stratigraphic bases to assign the Australian record to the early Tournaisian are rather weak (similarity of a vertebrate taxon with a late Devonian form and late Tournaisian dating on an orogenic event recorded in a different area), a general Tournaisian age seems more appropriate. So, all the conclusions proposed by the authors have to be rejected, especially the claim of first occurrence of reptile tracks in the early Tournaisian, an earlier footprint record of *Notalacerta* and the potential implication for the Molecular divergence dates. The authors should substantially improve the ichnologic study and focus on the implications of the first occurrence of *Hylopus hardingi* in the lower hemisphere which is potentially the earliest worldwide. This keeping in mind that this ichnotaxon is currently attributed to a wide range of early anamniotes. So, I suggest to reject this work, because the study is biased and no conclusion is supported by data.

Referee #3

(Remarks to the Author)

The authors provide significant new data bearing the age ad origins of major tetrapod clades of the Vertebrata. Though not in the form of body fossils, the ichnofossil data (in this case footprints) are compelling. They provide compelling evidence that the origin of the amniote stem must be pushed as deeply as the Frasnian or Fammenian (in their opinion the older Frasnian of the Devonian period. To suggest that this might be described as a tectonic change in our perceptions of major tetrapod evolution might be an exaggeration, but it significant – and in this reviewer's opinion, well significant enough to warrant publication.

The study documents new trackway evidence that is convincingly demonstrated as having been made by amniotes, specifically reptiles. I am convinced by the presentation, but would note that the concern that the fifth digit is not consistent in presentation might be attributed to variation in substrate pliability. (See the work of Gatsey ad Middleton as an example.) The evidence for being amniote, specifically reptiles, rests in part on the claw marks. Again this is agreeable, though they are a bit short on some references, particularly the work of Reisz and colleagues on the presence of claws and potentially horny claws as a vertebrate features. This however is a very minor concern.

The data set is small, so, statistics are not of concern, but while on the subject of measurements, it might be noted that the authors propose a body length for the suggested trackmaker as approximately 80 cm. That seems large at first glance, but data are data. I would however suggest they also include a speculation on snout-vent length of the trackmaker as opposed to only overall length including tail.

That said, I do have some minor suggestions, and one somewhat more substantive suggestion prior to potential publication:

Throughout the manuscript, they refer to the major amniote stem taxon Diadectomorpha, but in the abstract to diadectids specifically. As diadectids are generally considered the most derived of the diadectomorphs, I would suggest changing diadectids to diadectomorphs (line 129).

Line 147: While this may not be germane to this paper, the authors do suggest that depositional regimes are primarily lowland in nature. They should be awayre of the work of Berman, Sumida, Martens and colleagues of the Bromacker locality as evidence upland deposition in an intermontane paleograben (evidence of earlies terrestrial ecosystems, and also a very famous footprint locality).

The authors refer to both manus and pes imprints as "feet". Most would suggest the term foot should be restricted to the pes, and hand + feet as podal, or autopodal.

Reference 22; line 470. Capitalize 'paleozoic'.

My strongest suggestion/requirement for publication would be to properly update some of the geological data in the Supplementary material. In lines 87-93, they compare to ichnotaxa known from the early Permian section of north-central Texas, specifically referencing ichnotaxa of the "Choza Formation". Hentz (1988) and again Nelson et al (2013) have pointed out that the 'Choza' is not definably distinct from the Arroyo, Vale, and Choza Formations of Romer (1974) and Olson's (various) previous interpretations. Instead of "Choza" the authors would be better served as designating these taxa as upper Clear Fork Formation (uppermost Cisuralian/Leonardian). See references below:

Hentz, T. F. 1988. Lithostratigraphy and paleoenvironments of Upper Paleozoic continental red beds, north central Texas: Bowie (new) and Wichita (revised) groups. Bureau of Economic Geology: Report of Investigations, 170:55 pp.

Nelson, W.J., R.W. Hook, and D.S. Chaney. 2013. Lithostratigraphy of the Lower Permian (Leonardian) Clear Fork Formation of north-central Texas. New Mexico Museum of Natural History and Science, Bulletin 60:286–311.

Version 1:

Reviewer comments:

Referee #1

(Remarks to the Author)

I thank the authors for the care with which they addressed my previous comments and the extensive time spend incorporating changes in the MS and supplemental information regarding the dating of the Snowy Plains Formation and the proper placement of the ex situ slab from which the trackways were found. I think you have convincingly argued for a Tournaisian date, and towards the earlier side of the Age. I appreciate that the authors admit, in the absence of internal radiometric dates, that the age estimation "should definitely be regarded as an approximation, but it is clear that our trackway slab is no younger than mid-Tournaisian" and I am happy with that. However, in the MS abstract they still argue that the slab is "securely dated to the early Tournaisian (approximately 354-358.9 million years old)" which does not reflect the uncertainty they accept in the response to reviews. I'd be happier with the phrasing "Early in the Tournaisian"

This is a minor point overall for the paper, I admit, but this paper is important enough it can handle this real uncertainty. Unless the authors have other studies on rates of sedimentation (which can be remarkably fast) or durations of depositional hiatuses (which can mask over long times) of the members above and below the trackway slab that are providing the

brackets around the uncertainty, 354-358.9 seems an overly precise estimation. Moreover it misses out on the possible irony of a date of 353 MA, which would mean that you would have found the earliest crown amniote trackway while leaving the oldest crown tetrapod track date unchanged.

That aside, I'm fine with the submission as is, well done all, I look forward to its reception once published.

Referee #3

(Remarks to the Author)

This is now the second time I have reviewed this manuscript. I have had the opportunity to review the revised manuscript, with an eye toward whether I am satisfied with their responses to my own concerns, but I have also considered their responses to the other reviewers.

(1) I am satisfied with their responses to my (Reviewer 3) initial concerns. I do understand they are constrained by quoting of a previous reference, and that they therefore cannot reference the updated north-central Texas sequence correctly. However, it remains misleading, and I suggest adding a sentence along the lines of: "The more recently revised north-central Texas sequence was revised by Hentz and colleagues (reference), but these revisions do not alter/impact the conclusions presented here."

(2) I agree with the response to Reviewer 1 that it is not necessary to address the position of Lucas (2015). Lucas made numerous broad ranging conclusions in that publication, a number of which were not supported by actual examination of specimens or data. Thus, I feel the authors are in a reasonable position not to consider that particular "study".

Manuscript 2024-05-09700: Response to Referees

We thank the referees for their thorough appraisal of our manuscript. There are two obvious questions to raise in relation to this work, namely the age and the trackmaker identity of the footprints; we are pleased that the referees have addressed both of these and we are confident that we can provide satisfactory answers to their criticisms. This does, of course, mean that we emphatically reject Referee #2's alternative interpretation of our footprints as belonging to the non-amniote ichnogenus *Hylopus*. We will explain in detail, with illustrations, why we disagree with this identification.

Referees' comments are shown in italics, our replies in plain text.

Referee #1 (Remarks to the Author):

This is an interesting MS on a set of trackways discovered in Australia, and the implications of those trackways for our understanding of tetrapod evolution if correctly identified. It is original in its presented data, and extremely important if correct. Data and methodology are correctly presented. Statistics were not applied in the paper.

Reply: Thank you!

It seems to me the critical point upon which everything hangs is the date of the trackway. Its somewhat concerning the blocks upon which the tracks were found was loose. The authors pass this off as "essentially in situ" which makes me scratch my head a little. What is the degree of offset of this block from the main intact bed? Millimeters? Centimeters? Meters? Does this block fit into place on the main bedrock?

Reply: This and the following two paragraphs really form a single multi-part query and are best answered together. See response after the third paragraph.

That the area is structurally complex makes dating this slab fraught with difficulty. There are no radiometric dates. Argumentation, elaborated upon to a degree in Supplemental Information 2, is relative based on two factors: presence or absence of key taxa, and an assumption of the source of sedimentation. If correct that the source of sedimentation is the Kanimblan Orogeny, if it was of short duration, as argued by others, and if dating in the basin in which the trackways were found is the same as in the basin with similar lithologies that could be dated, then the date is constrained to be between the Devonian-Carboniferous Boundary and the end of the Tournaisian. How the authors settled on a date of the earliest Tournaisian (which would further make this locality unique) I have no strong idea. But that is a long list of nested assumptions and it is upon this that everything else hangs--which is a range extension of amniote trackways back in time by 20 odd million years. I would prefer the authors be a bit more circumspect with respect to the real uncertainty that is being passed over, because doing so doesn't fundamentally change the nature of their analysis for the impact on tetrapod evolution. On the other hand, should this slab be from a younger depositional event, then the impact of the paper decreases proportionately to the shortening of the possible range extension.

Relatedly, I pose a question to the authors: can a similar analysis be performed that rules out a date of late Visean to early Serpukhovian can be ruled out if the identification of the sediment source (uplift due to the Kanimblan Orogeny) doesn't hold? Are there deposits of those ages within this "structurally complex region" present? I think their argument would be bolstered if no deposits of this age or especially younger (say, Visean) were anywhere close to the discovery locality of the specimen. A lot is described about older sediments, in other words, but not much about younger. Please fill out the stratigraphic profile to help the reader agree with your estimates.

Reply: We provided a fairly comprehensive account of the regional geology in our *Supplementary Information 2* file, but the main points bear repeating here, and of course they are absolutely central to the argument of this manuscript. While it is true that the Lachlan Fold Belt is structurally complex, it is also well-studied and is not overly difficult to interpret. It is not entirely reasonable to refer to the existing knowledge base, built up over more than a century of geological exploration under the auspices of the Geological Survey of Victoria and others, as "nested assumptions". Maps and stratigraphic columns from the Geological Survey of Victoria are attached to the end of this Response.

As we explained in *Supplementary Information 2*, the Lachlan Fold Belt contains a number of sedimentary basins that yield characteristic Late Devonian fossil assemblages containing index taxa such as placoderms, tristichopterids and holoptychiids. The Mansfield Basin, however, does not. The sedimentary succession in this basin, known as the Mansfield Group, starts with the Mt. Kent (or Timbertop) Conglomerate, which is overlain by the Snowy Plains Formation (SPF) comprising just under 2km thickness of sandstones and siltstones. Our trackway slab comes from the Home Station Sandstone member, roughly in the middle of the SPF. A single pebble with a placoderm bone in it has turned up in the Mt. Kent Conglomerate, indicating the the source of the clasts was a Late Devonian sedimentary rock, but apart from that the vertebrate fossil assemblage of the Mansfield Group is completely devoid of Devonian index taxa. What it does contain is chondrichthyans (*Ageleodus*, *Ctenacanthus*), acanthodians (*Gyracanthides*, *Acanthodes*), actinopterygians (*Mansfieldiscus*, *Novogonatodus*), and a rhizodont sarcopterygian (*Barameda*). All are groups that cross the Devonian-Carboniferous boundary, and would thus in principle allow a Devonian age for the SPF. However, the complete absence of placoderms, tristichopterids and holoptychiids, all of which are known elsewhere to disappear at the Devonian-Carboniferous boundary (probably victims of the Hangenberg Crisis), suggests an Early Carboniferous date is most likely. On the Geological Survey of Victoria map from 1997, the Mansfield Group is interpreted as Late Devonian, but in the 2006 version it has been reassigned to the Early Carboniferous (see figures at end of Response).

Before moving on to consider the youngest possible age for the slab (which is central to the Referee's argumentation, and indeed to ours), we need to address the question of how securely the slab can be assigned to the Home Station Sandstone member, given that it was found as a loose block. Could it have been transported from a different stratigraphic unit? To answer that, let's look at the site itself:

[FIGURE REDACTED]

The photo on the right, taken by John Eason, shows the spot where he and Craig Eury found the slab, and where lead author Prof John Long, a trained geologist, was taken to by Craig to examine the outcrop. On the left is an image from Google Earth; the find spot was approximately where you see the words "Broken River". As you can see this is a rather low-relief landscape, with bedrock only outcropping along the river. Apart from recent alluvium, there's nothing but SPF (named "Devil's Plain Formation" on the 1997 map, the terminology has changed since then) for several kilometres in any direction from the find site. Upstream – the only direction from which the slab could conceivably have travelled – the river eventually crosses over onto Quarternary sands and gravels of obviously recent origin. The slab itself is pristine, showing no evidence of long-distance transport, and is similar in lithology to the local outcrop. There is thus no way the slab could realistically have come from anywhere other than the Home Station Sandstone member of the SPF. John Long examined the site and concluded that it had spalled off from the local outcrop by surface weathering and soil action.

Another point further supports this conclusion. Australia was moving rapidly southwards into colder latitudes during the Early Carboniferous, as shown by recent palaeomagnetic work (Klootwijk 2009, 2010). This is reflected by a dramatic change in the sedimentation, from the SPF which was evidently deposited in a fairly warm environment, to the next youngest sedimentary formations of the region, the Upper Carboniferous Wild Duck Formation and Boorhaman Conglomerate, which postdate the Kanimblan Orogeny. These are periglacial deposits containing tillites and diamictites, completely unlike the SPF and clearly not the source of the slab. In any case the nearest outcrops of these formations are many kilometres away. We are thus confident in assigning our slab to the Home Station Sandstone member of the Snowy Plains Formation. We are also confident that the SPF as a whole predates the Kanimblan Orogeny; this is the consensus view of all structural geologists who have worked in the area, reflected in the maps and stratigraphies published by the Geological Survey of Victoria, and we have no reason to doubt that it is correct.

This, then, leaves the question of the exact age of the Kanimblan Orogeny. As we explained in *Supplementary Information 2*, the late phase of the orogeny saw the emplacement of a series of granites with radiometric dates ranging between 341-314 million years, plus an early outlier at 358 million years. Even a relatively brief orogeny takes many millions of years to complete, and it is clear from the regional geology that main phase of the orogeny occurred after the cessation of (indeed, terminated) sedimentation in the Mansfield Basin. Thus, even if we disregard the early outlier granite date the beginning of the orogeny and the end of SPF deposition must be considerably older than 341 million years. The Geological Survey of Victoria places the onset of the orogeny at 350 million years, giving an

approximate age of 355 million years for the Home Station Sandstone member of the SPF. This should definitely be regarded as an approximation, but it is clear that our trackway slab is no younger than mid-Tournaisian. A post-Viséan date is absolutely out of the question: it would require a wholesale rejection of the entire established understanding of the regional geology of Victoria, and from a palaeobiological perspective would require us to explain how these early amniotes could live in the frigid wasteland by the very edge of the south polar ice sheet.

We have expanded Supplementary Information 2 to include an explanation (illustrated with the locality photo) of why we are confident that the trackway slab originated locally.

One other point that can affect date constraints for the discussion of tetrapod evolution is over the validity of the Zachełmie trackways, which is disputed by some authors (Lucas 2015). This is a more minor point because that critic accepts the Valentia trackways, so its a slight younger adjustment of the origination minimum age estimate. I accept that some of the authors are deeply involved in this debate but this point passes without even a mention; I think it should warrant at least a sentence or two of how that would shift estimates of ancient tetrapod origins.

Reply: There is no "dispute" worthy of the name. Lucas has never examined the Zachełmie tracks in person, whereas we have worked extensively with them (both those published by Niedzwiedzki et al. in 2010 and others discovered subsequently), and we remain confident in our interpretation of them as tetrapod tracks. Lucas 2015 substantially mischaracterises the tracks, and his interpretation cannot explain their characteristics such as digit impressions and occurrence in obvious linear trackways. In any case, as the Referee notes, the impact on the minimum age of limbed tetrapods is fairly minimal (the Valentia Island tracks, which are accepted as tetrapod tracks by Lucas, are only about 5 million years younger than Zachełmie) and does not affect the core argument of the present manuscript at all. We would prefer to not cite Lucas 2015; it is not really relevant to the argument, explaining what's wrong with it would require at least a paragraph, and a briefer mention would lend it a sheen of credibility that it frankly does not deserve.

On the overall conclusion of the paper: I think the evidence has been building for some time that the tetrapod fossil record is missing considerable representation, with recent first occurrences for colosteids and crassigyrinds being dragged down in the record some 20-30 million years into the earliest Carboniferous and possibly into the Devonian for the former. Are we sampling shifting facies not overall tetrapod evolution? Is this why these amniote prints are appearing in Gondwana earlier than seen in Laurasia as argued by the authors? All good points to get into the literature and the field as a whole discussing more actively.

Reply: We very much agree with these sentiments. The early tetrapod record is evidently far less complete than we have tended to believe, very likely because of facies and/or biogeographical 'filtering'. It is as yet too early to say exactly what factors have been at play, but at least we are beginning to assemble some pieces of evidence.

To improve the presentation of the paper: I think my biggest recommendation to the authors is to be very very careful with their language around "early" and "late", because it can be

confusing to a general audience when what is meant shifts depending on perspective. One thing about time from the recent back, but about rocks from the bottom up, and what is meant by the use of modifiers can suggest to the reader the opposite of what is meant by the authors. For example, on line 173, after having discussed amniotes originating in both trackways and body fossils in the Serpukhovian, the authors say their new trackways show that "these dates are substantially too late". It caused me, a fellow paleontologist, some pause to figure out what was meant was that in the rock record those old ideas about amniote origins postdate the implied dates by the new fossils (i.e. thinking about data provided by ROCKS), which are from closer to the Devonian. When I first read it though, I was thinking about TIME since dates were being discussed, and a late date would be OLDER (i.e., further back in time). I suggest in that instance to consider rephrasing to "[those dates are] substantially too recent" or "substantial underestimates". Especially for the generalist audience of Nature you should triple check all of these instances to ensure what is clear in your mind is being communicated to folks who don't live and breathe both time and rocks like we paleontologists do (for good or ill).

Normally at this point I would pass the rest of the minor needed edits to a marked up MS copy (which I still attach) but Nature wants for everything to be written out here so, for your edification, the fine points:

Reply: Thank you, we consider ourselves edified!

- Line 176: strike the comma after Poland

Reply: we went for "Silesia in Poland", without comma

- Line 176: data show (data is plural)

Reply: corrected

- Line 177: earlier vs older. Do they occur closer to recent or further back in time?

Reply: Here we find ourselves at a loss, because we don't entirely understand why this presents a problem. "Earlier" and "older" are functionally synonymous. "Older" embodies a static perspective looking back from the present, "earlier" sees time as a series of events. For example, it is true that Tournaisian sediments are older than Viséan sediments, and it is true that Tournaisian sediments were laid down earlier than Viséan sediments. "Earlier" and "older" both refer to events further back in time. We have left the text unchanged for now but are happy to refer this question to the Editor for adjudication.

- Line 222-223: a pedantic point--you aren't looking at skin, but skin impressions into substrate. Substrate can lack resolution to preserve finer details of soft tissues.

Reply: "impression of the" has been added

- Line 230-231: Xenopus has claws. These trackways could of course be made by a yet unknown group that acquired amniote-like claws in parallel to amniotes. I understand the authors are implicitly making an argument by parsimony here but since one of the points of the paper is just how poorly taxa are represented by body fossils at this time in the rock record missing out on a group is certainly conceivable.

Reply: A fair, and important, point. We have expanded this section to mention *Xenopus* (and a few other taxa) and in general make the discussion more nuanced.

No further comments. The paper is very well written and thought provoking; well done!

Reply: Many thanks!

Referee #2 (Remarks to the Author):

Comment: Referee #2's comments reached us as a single block of continuous text. We have broken it up into paragraphs to make reading and responding to it more manageable.

This work reports, describes and interprets the first discovery of tetrapod footprints from the Snowy Plains Formation of Australia. It is unquestionably a new fossil record of tetrapod footprints. The material is well-illustrated, although the footprint outline is not well-traced and some key features such as a false-color scale bar for the 3D models are not provided.

Reply: The false-colour scale bars of the 3D models were lost when the images were cropped to fit into the multi-panel figure. We would prefer to not add them as inserts on the figure, which would force us to reduce the size of the actual footprint images, but if the Editor so desires we can create a supplementary file containing the uncropped images. Rather than trace the footprints in great detail, a step that inevitably introduces a degree of subjectivity, we prefer to rely on actual images of the prints such as the false-colour 3D models.

In the description, some important features such as the preservation as convex hyporelief or concave epirelief are not specified.

Reply: The prints are preserved as concave epirelief. This is now stated in the text.

Also, the specimen is described as sandstone although the surface is clearly finer.

Reply: The rock slab represents a fine-grained, silty sandstone. We did not perform additional petrographic studies, the identification is based on macroscopic observations.

Most important, these footprints are compared to the earliest ichnotaxa attributed to amniotes such as *Notolacerta*, *Dromopus*, *Varanopus* and *Dimetropus* and a similarity with *Varanopus* is noticed. However, this work surprisingly does not discuss possible differences and similarities of the material with *Hylopus*, the most common anamniote tracks of the Mississippian. Material currently assigned to this ichnogenus in some work was mistaken for a reptile track (e.g., Falcon-Lang et al., 2007, 2010) and subsequently consistently and thoroughly attributed to anamniotes (e.g., Keighley et al., 2008; Fillmore et al., 2012; Lucas, 2019; Marchetti et al., 2019, 2021). The described material can be confidently assigned to *Hylopus hardingi*, based on a well-preserved pes (Fig. 2b-c) and manus (Fig. 2e-f). Diagnostic features include, among others, a well-impressed basal pad of digit I in the pes, a clearly more ectaxonic manus imprint compared to the pes, a relatively long digit V in the pes, the non-overlapping digit bases. Trackway B can only tentatively assigned to *Hylopus* because of its poor preservation due to incompleteness and digit tip dragging. The purported claw traces are mostly digit tip drag traces, which occur often in the Palaeozoic anamniote track record and do not imply the presence of claws (e.g., Lucas et al., 2022). Also, the potential presence of pointy digit tips/claws in early anamniotes cannot be excluded, also considering the occurrence of claws in some modern anamniotes (e.g., Maddin et al. 2007). Moreover, the comparison with *Varanopus* is made based on outdated literature (e.g., Haubold and Lucas, 2003, Voigt, 2005). *Varanopus* is clearly different from the described material because of the clearly overlapping digit bases, the more ectaxonic pes imprint, the concave proximal sole/palm margin and the possible occurrence of claw traces.

Reply: We agree with a few aspects of this, but emphatically disagree with others – and we will argue below that the points of disagreement are critically important to evaluating the phylogenetic position of the track maker.

We agree that the footprints in Trackway A show a few shape similarities with prints assigned to the ichnotaxon *Hylopus hardingi*. There is a modest basal pad by digit I, though we would not call it "well-impressed"; it is only a slight bulge, nothing like the distinct knob seen in classic *Hylopus hardingi* (e.g. Marchetti et al. 2021, Figure 1D, 2D). The digit impressions do not overlap basally. Overall there is a sense that the sole is slightly wider, and the trackmaker perhaps somewhat less digitigrade, than in *Varanopus*, *Notolacerta* and *Dromopus*, although it is difficult to know how much of this is due to the specific character of the substrate and its response to the feet of the trackmaker. We will return below to the meaning of these differences. Conversely, one of the diagnostic features of *Hylopus hardingi* is the short digit V in the manus, which is described as being "as long as digit II" (Marchetti et al. 2021, p. 9). In the Australian manus prints, digit V is much longer, approximately the same length as digit III. This in itself falsifies the claim that our tracks can be "confidently assigned to *Hylopus hardingi*". That conclusion is further reinforced by the presence of bona fide claw prints in our footprints (see below), but in fact we believe the problem is more fundamental: "*Hylopus hardingi*" as currently defined does not even approximate to a biotaxon, it is simply a shape category, and moreover one which has turned into something of a dustbin.

The problem is exemplified by the revised diagnosis of the ichnospecies (and -genus) presented by Marchetti et al. 2021, which contains the following statement about the digit tips: "*Digit terminations tapering or enlarged, rarely with pointed end*". This admits

trackmaker feet with morphologically different digit ends – enlarged, tapering or with some kind of pointed and presumably keratinised tip – into a single ichnospecies. However, we know from living animals that the character of the toe tips is not phylogenetically labile and certainly not individually variable; for example, claws are almost invariably present in amniotes but absent (with a few scattered exceptions such as *Xenopus* and *Onychodactylus*) in amphibians. It is thus clear that "*Hylopus hardingi*" so defined must represent footprints of a range of different tetrapods, not all of which are necessarily closely related. Indeed, if the "*Hylopus hardingi*" footprints in Marchetti et al. 2021 Figure 1D, 2D are compared with those in Figure 4I, phylogenetically significant differences are immediately visible: ball-shaped digit terminations and a knob-shaped basal pad by digit I in Figure 1D, versus pointed and hooked digit terminations (possibly claws) and no basal pad by digit I in Figure 4I. These are almost certainly separate biospecies, and we would not rule out an amniote identification for Figure 4I. The uniting features of *Hylopus hardingi* cited in the definition may simply be symplesiomorphies of a range of stem (and basal crown?) amniotes, a point that we will return to below. "Assigning" a tetrapod trackway to *Hylopus hardingi* as currently defined is thus not a phylogenetically meaningful statement, except in a very broad and vague sense.

This brings us back to the question of claws. Claws are unquestionably a derived character within Tetrapoda and thus potentially phylogenetically informative. They are present in crown amniotes, and convergently in some few extant amphibians. They appear to have been present in microsaurians, but the significance of this is uncertain as there is no consensus about the phylogenetic position of this group and some analyses recover them as crown amniotes. It has been argued that they are present in diadectids, based on the surface structure of the terminal phalanges, but the footprint record doesn't bear this out (Voigt et al. 2007). In short, claws are a derived character that will tend to assign the trackmaker to crown Amniota, although convergent evolution of claws in early non-amniote tetrapods should be given consideration as an alternative possibility if the 'amniote signal' is contradicted by other aspects of the footprint morphology.

We maintain, contrary to the claim of Referee #2, that the Australian footprints show perfectly good claw impressions, indistinguishable from those of sauropsid ichnotaxa such as *Notolacerta* and *Varanopus*. They are certainly not "*mostly digit tip drag traces*". The two trackways and isolated pes print on the slab conveniently show us three different styles of claw-mark preservation, created by different sediment-foot interactions (rather dry surface before the rain for the isolated pes print, soft surface after the rain for Trackway A, somewhat drier surface for Trackway B). In the isolated pes print, the claw marks are deflected medially by about 90 degrees, with a distinct angle between claw mark and digit. This implies that the claw tips were nearly vertical (in the same manner as in a modern lizard) and that the distal parts of the digits twisted so that the claws lay down on their sides when stepping on an unyielding substrate. In Trackway A, made on softer ground, the claws have not flipped sideways to the same degree; instead they have left sharp, narrow marks, clearly distinct and separate from the soft fleshy digit. Finally, in Trackway B, the claws have made long scratches, but the sharp separation between claw and digit impression can still be clearly seen. These are all distinctive characteristics of genuine claw impressions.

The presence of claw impressions rules out identification of the Australian footprints with the classic *Hylopus hardingi* as represented by Marchetti et al. 2021 Figure 1D, 2D (and many other specimens). Quite apart from the smaller digit V on the manus, and the presence of a prominent knob-shaped pad at the base of digit I, classic *Hylopus hardingi* differs from the Australian tracks in having bulbous, rounded toe tips without claws. Whether claw

impressions may be present in other material assigned to *Hylopus*, as is possibly the case (see Marchetti et al. 2021 Figure 4I), is irrelevant; if so, it just shows either a) that the specimens have been assigned in error, or b) that “*Hylopus hardingi*” is a dustbin taxon based on morphometric similarity without phylogenetic content, or both.

So, what does all this mean for the phylogenetic position of the Australian footprints? They have claw impressions, and a general ectaxonic foot morphology very similar to that of acknowledged early sauropsid footprints such as *Notalacerta*, *Varanopus* and *Dromopus*. However, they combine this with a cluster of subtle, somewhat *Hylopus*-like characteristics – slightly wider separation of the digit bases, a modest basal pad by digit I, and an overall sense of being less digitigrade – that are probably plesiomorphies shared with stem amniotes. The conclusion is obvious: the trackmaker was a very primitive amniote, either a sauropsid basal to *Notalacerta*, *Varanopus* and *Dromopus*, or else a taxon close to the amniote crown group node. No part of the character complement suggests a placement in another region of tetrapod phylogeny. This represents a slight modification of our previous position (that the Australian taxon was specifically very close to *Varanopus*) and is reflected in a modification of our text. However, the essentials of our discovery and its implications remain unchanged. Here’s a straightforward comparative figure of the isolated pes print from the Australian slab, together with examples of *Varanopus* and *Hylopus*, to illustrate some of these points. Note particularly the strange ball-shaped toe tips in *Hylopus*, contrasting with the claw impressions in *Varanopus* and the Australian footprint.

[FIGURE REDACTED]

*Also, the isolated footprints from Poland, of which the locality is not provided, are assignable to *Hylopus hardingi* manual imprints because of their marked ectaxony and overall digit proportions and morphology.*

Reply: No, they are not. We find this claim quite surprising, as even a cursory comparison shows numerous differences between the two. Below, we show *Notolacerta* from Silesia alongside a classic *Notolacerta* from the Pennsylvanian of the USA and *Hylopus hardingi* from the Mauch Chunk Formation. The distribution of similarities and differences should be obvious: the Silesian footprints are not *Hylopus*. The suggestion that these are manual imprints is also an error; the figured footprints are pes prints, but the Wałbrzych Formation collection contains both manus and pes impressions. In all studied manus and pes specimens from the Wałbrzych Formation digit V is very long, similar to *Notolacerta*, contrasting with *Hylopus hardingi* in which digit V of the manus is very short (see above). Even on the basis of this observation alone, an assignment of the Wałbrzych Formation to *Hylopus hardingi* can be ruled out, but there are plenty of other differences as well – notably, the presence of claw impressions and the absence of both ball-shaped toe tips and a basal pad by digit I. In short, the Wałbrzych Formation prints have nothing at all in common with *Hylopus hardingi* beyond being pentadactyl and ectaxonic.

[FIGURE REDACTED]

It can be added that the Silesian footprints are part of an extensive ichnoassemblage, also including real *Hylopus* footprints, which one of us (G.N.) is in the process of describing. Full locality data and details about other studied tracksites from the Silesia region are now provided in a new Supplementary Information file, *Supplementary Information 4*.

Finally, the stratigraphic bases to assign the Australian record to the early Tournaisian are rather weak (similarity of a vertebrate taxon with a late Devonian form and late Tournaisian dating on an orogenic event recorded in a different area), a general Tournaisian age seems more appropriate.

Reply: See response to Referee #1. An early Tournaisian age is well substantiated.

*So, all the conclusions proposed by the authors have to be rejected, especially the claim of first occurrence of reptile tracks in the early Tournaisian, an earlier footprint record of Notalacerta and the potential implication for the Molecular divergence dates. The authors should substantially improve the ichnologic study and focus on the implications of the first occurrence of *Hylopus hardingi* in the lower hemisphere which is potentially the earliest worldwide. This keeping in mind that this ichnotaxon is currently attributed to a wide range of early anamniotes. So, I suggest to reject this work, because the study is biased and no conclusion is supported by data.*

Reply: We disagree with this assessment.

Referee #3 (Remarks to the Author):

The authors provide significant new data bearing the age ad origins of major tetrapod clades of the Vertebrata. Though not in the form of body fossils, the ichnofossil data (in this case footprints) are compelling. They provide compelling evidence that the origin of the amniote stem must be pushed as deeply as the Frasnian or Fammenian (in their opinion the older Frasnian of the Devonian period. To suggest that this might be described as a tectonic change in our perceptions of major tetrapod evolution might be an exaggeration, but it significant – and in this reviewer’s opinion, well significant enough to warrant publication.

Reply: Thank you!

The study documents new trackway evidence that is convincingly demonstrated as having been made by amniotes, specifically reptiles. I am convinced by the presentation, but would note that the concern that the fifth digit is not consistent in presentation might be attributed to variation in substrate pliability. (See the work of Gatsey ad Middleton as an example.) The evidence for being amniote, specifically reptiles, rests in part on the claw marks. Again this is agreeable, though they are a bit short on some references, particularly the work of Reisz and colleagues on the presence of claws and potentially horny claws as a vertebrate features. This however is a very minor concern.

Reply: Thank you for this positive assessment. We have expanded the discussion of claws in tetrapods.

The data set is small, so, statistics are not of concern, but while on the subject of measurements, it might be noted that the authors propose a body length for the suggested

trackmaker as approximately 80 cm. That seems large at first glance, but data are data. I would however suggest they also include a speculation on snout-vent length of the trackmaker as opposed to only overall length including tail.

Reply: It is, of course, impossible to accurately infer the total length of the animal because of uncertainties regarding neck and (in particular) tail length. However, we wanted to give a reader unfamiliar with trackway evidence some sort of ballpark mental image of the size of the trackmaker. *Varanus salvator* is a rather long and thin lizard with a long tail; applying a shorter and chunkier model would give a shorter total length. We do give a hip-shoulder distance, which is quite well-constrained. A snout-vent estimate, on the other hand, is only slightly more reliable than an overall estimate, because the length of the neck and head are unknown and can vary quite a lot; we don't feel that including such an estimate would add very much to the manuscript.

That said, I do have some minor suggestions, and one somewhat more substantive suggestion prior to potential publication:

Throughout the manuscript, they refer to the major amniote stem taxon Diadectomorpha, but in the abstract to diadectids specifically. As diadectids are generally considered the most derived of the diadectomorphs, I would suggest changing diadectids to diadectomorphs (line 129).

Reply: Agreed and implemented, except where we speak of "diadectids and limnoscelids" which both lie within the diadectomorphs.

Line 147: While this may not be germane to this paper, the authors do suggest that depositional regimes are primarily lowland in nature. They should be aware of the work of Berman, Sumida, Martens and colleagues of the Bromacker locality as evidence upland deposition in an intermontane paleogaben (evidence of earliest terrestrial ecosystems, and also a very famous footprint locality).

Reply: Fair point. The text has been modified. Incidentally, the Late Devonian basin of East Greenland, where we are currently engaged in fieldwork and finding new tetrapods, is also an inland basin located about 1000 km from the nearest sea.

The authors refer to both manus and pes imprints as "feet". Most would suggest the term foot should be restricted to the pes, and hand + feet as podal, or autopodal.

Reply: We don't dispute that this would be technically correct, but we are nevertheless in two minds about it. In a journal like *Nature*, many of the readers will be non-specialists, and we are inclined where possible to use a terminology that's close to vernacular usage and therefore easy to understand. In vernacular terms, the autopods of a lizard or similar animal would all be described as "feet", with the manus specified as "front foot" and the pes as "hind foot". The manus would typically only be described as a "hand" if it is a grasping structure

used for manipulating objects, as in primates. Describing fossil footprints as “podal (or autopodal) prints” would, we fear, mystify many readers and make the paper less easy to understand. However, we are happy to defer to the judgement of the Editor on this point.

Reference 22; line 470. Capitalize ‘paleozoic’.

Reply: Done.

My strongest suggestion/requirement for publication would be to properly update some of the geological data in the Supplementary material. In lines 87-93, they compare to ichnotaxa known from the early Permian section of north-central Texas, specifically referencing ichnotaxa of the “Choza Formation”. Hentz (1988) and again Nelson et al (2013) have pointed out that the ‘Choza’ is not definably distinct from the Arroyo, Vale, and Choza Formations of Romer (1974) and Olson’s (various) previous interpretations. Instead of “Choza” the authors would be better served as designating these taxa as upper Clear Fork Formation (uppermost Cisuralian/Leonardian). See references below:

Hentz, T. F. 1988. Lithostratigraphy and paleoenvironments of Upper Paleozoic continental red beds, north central Texas: Bowie (new) and Wichita (revised) groups. Bureau of Economic Geology: Report of Investigations, 170:55 pp.

Nelson, W.J., R.W. Hook, and D.S. Chaney. 2013. Lithostratigraphy of the Lower Permian (Leonardian) Clear Fork Formation of north-central Texas. New Mexico Museum of Natural History and Science, Bulletin 60:286–311.

Reply: We appreciate this additional information, but as the text in question is a direct quote from Haubold & Lucas 2003, identified as such and placed within double quotation marks “”, we are unfortunately unable to change it.

[FIGURE REDACTED]

[FIGURE REDACTED]

[FIGURE REDACTED]

Second review (we have not been asked to respond to these comments, but have implemented the referees' suggestions):

Referee #1 (Remarks to the Author):

I thank the authors for the care with which they addressed my previous comments and the extensive time spend incorporating changes in the MS and supplemental information regarding the dating of the Snowy Plains Formation and the proper placement of the ex situ slab from which the trackways were found. I think you have convincingly argued for a Tournaisian date, and towards the earlier side of the Age. I appreciate that the authors admit, in the absence of internal radiometric dates, that the age estimation "should definitely be regarded as an approximation, but it is clear that our trackway slab is no younger than mid-Tournaisian" and I am happy with that. However, in the MS abstract they still argue that the slab is "securely dated to the early Tournaisian (approximately 354-358.9 million years old)" which does not reflect the uncertainty they accept in the response to reviews. I'd be happier with the phrasing "Early in the Tournaisian".

This is a minor point overall for the paper, I admit, but this paper is important enough it can handle this real uncertainty. Unless the authors have other studies on rates of sedimentation (which can be remarkably fast) or durations of depositional hiatuses (which can mask over long times) of the members above and below the trackway slab that are providing the brackets around the uncertainty, 354-358.9 seems an overly precise estimation. Moreover it misses out on the possible irony of a date of 353 MA, which would mean that you would have found the earliest crown amniote trackway while leaving the oldest crown tetrapod track date unchanged.

That aside, I'm fine with the submission as is, well done all, I look forward to its reception once published.

Referee #2:

This referee declined to review the manuscript a second time.

Referee #3 (Remarks to the Author):

This is now the second time I have reviewed this manuscript. I have had the opportunity to review the revised manuscript, with an eye toward whether I am satisfied with their responses to my own concerns, but I have also considered their responses to the other reviewers.

(1) I am satisfied with their responses to my (Reviewer 3) initial concerns. I do understand they are constrained by quoting of a previous reference, and that they therefore cannot reference the updated north-central Texas sequence correctly. However, it remains misleading, and I suggest adding a sentence along the lines of: "The more recently revised north-central Texas sequence was revised by Hentz and colleagues (reference), but these revisions do not alter/impact the conclusions presented here."

(2) I agree with the response to Reviewer 1 that it is not necessary to address the position of Lucas (2015). Lucas made numerous broad ranging conclusions in that publication, a number of which were not supported by actual examination of specimens or data. Thus, I feel the authors are in a reasonable position to not consider that particular "study".